# A Preliminary Evaluation of the Cultural Appropriateness of the Tuning in to Kids Parenting Program in Germany, Turkey, Iran and China

**DOI:** 10.3390/ijerph191610321

**Published:** 2022-08-19

**Authors:** Sophie S. Havighurst, Rachel Choy, Ayca Ulker, Nantje Otterpohl, Fateme Aghaie Meybodi, Forough Edrissi, Chen Qiu, Kathy Kar-man Shum, Alessandra Radovini, Dana A. Hosn, Christiane E. Kehoe

**Affiliations:** 1Mindful: Centre for Training and Research in Developmental Health, Department of Psychiatry, The University of Melbourne, Melbourne, VIC 3010, Australia; 2Department of Early Childhood Education, Hacettepe University, Ankara, Turkey; 3Department of Psychology and Sport, Justus Liebig University of Giessen, Giessen, Germany; 4Research Center of Addiction and Behavioral Sciences, Shahid Sadoughi University of Medical Sciences, Yazd, Iran; 5Department of Clinical Psychology, Shahid Beheshti University of Medical Sciences, Tehran, Iran; 6Department of Psychology, The University of Hong Kong, Hong Kong; 7West China School of Nursing, Sichuan University, Chengdu 610017, China

**Keywords:** emotion socialization, emotion coaching, parenting, culture, intervention, adaptation, Tuning in to Kids

## Abstract

*Background*: Parenting interventions based on emotion socialization (ES) theory offer an important theoretically driven approach to improve children’s emotional competence and behavioral functioning. Whether such approaches are effective in different cultural contexts, and whether the methods of delivery used are appropriate and acceptable, is an important empirical question. This paper reports on the preliminary evaluation of an ES parenting intervention, Tuning in to Kids (TIK), in Germany, Turkey, Iran, and China. Pilot studies of TIK have been conducted in each country with mothers of 4–6-year-old children. *Method:* The current study used qualitative methods with thematic analysis to explore the cultural appropriateness of the program in each site. *Results:* Culture-specific challenges were found across all sites in changing parents’ beliefs about the value of encouraging children’s emotional expression and supportive emotion discussions. Emotion literacy of parents depended on their access to emotion terms in their language, but also to parents’ experiences with emotions in their family of origin and culture-related beliefs about emotions. Adaptations were required to slow the speed of delivery, to address issues of trust with parents in seeking help, and to provide more opportunities to practice the skills and integrate different beliefs about parenting. *Conclusion*: While this ES parenting intervention has been developed in a Western cultural context, slight adaptations to the delivery methods (rather than change to the content) appeared to contribute to cultural appropriateness. The next step will be to quantitatively evaluate these adaptations of TIK in the different countries using randomized controlled studies.

## 1. Introduction

Tuning in to Kids (TIK; [1]) is an Australian group-parenting program that teaches parents skills in emotion awareness, emotion regulation, and emotion coaching to facilitate supportive emotion socialization (ES) and promote children’s healthy emotional development [2,3]. The intervention has been found effective in improving children’s emotional and behavioral functioning in Australian studies [4,5,6], and evaluation trials are now being conducted in other countries around the world [7,8,9,10,11,12,13]. Research has shown that the way parents socialize emotions is culturally determined, with race, ethnicity, culture, and experience all contributing to parents’ responses to children’s emotions [2,14]. Examining whether the skills taught in TIK generalize across cultures thus requires investigation. Of interest was whether modifications to components of TIK were required to address widely differing culture-specific parental emotion-related beliefs and practices to facilitate culturally appropriate delivery of the program. This study reports on preliminary evaluations of the TIK program conducted with parents of preschool children in four diverse cultural contexts—Germany, Turkey, Iran, and China—by looking at themes from a qualitative survey completed by researchers who delivered the program in each country.

### 1.1. Parent Emotion Socialization

From early in life, children learn skills in understanding and managing their emotions, known as emotional competence [3]. These skills influence their social relationships, behavior, and capacity to direct attention and learn, and affect how they cope with stressful events and transitions, a finding seen across cultures (ibid). Children’s age and temperament partly determines emotional competence; however, parenting also shapes these skills [2,15]. When parents have effective emotional awareness and regulation, engage in supportive reactions to their children’s emotions, and assist their children with opportunities to understand and regulate emotions, it assists children to learn these critical skills (ibid). Parental reactions and discussion around children’s emotions that include validating the child’s emotions and scaffolding emotional understanding and regulation has been called emotion coaching [16]. Emotion coaching involves a parent (1) noticing the child’s emotion, (2) viewing it as an opportunity for connection and teaching, (3) helping the child name the emotion, (4) validating emotions with empathy, and (5) guiding the child to manage emotions and solve problems while setting limits around behavior. When parents regulate their own emotions in socially appropriate ways and use emotion coaching to connect and teach, it facilitates the development of children’s emotional competence and prosocial behavioral adjustment [2,15,16,17,18,19]. This occurs because children learn to pay attention and put words to their emotions and, through a process of scaffolding by their parents, gradually increase their ability to recognize and manage emotions across a range of contexts and experiences. In contrast, when parents are more emotionally dysregulated and emotionally dismissive of their children’s emotions (characterized by non-acceptance of emotions and responses that discourage, punish, or neglect emotions), poorer emotion competence and more behavior problems have been found [20].

### 1.2. Parenting and Emotion Socialization across Cultures

Although emotions are universal [21], the experience and expression of emotions varies according to the cultural display rules about emotions [2,3]. Individuals learn to express (or conversely avoid or suppress) emotions in accordance with emotion-related cultural norms and values [22]. Three main cultural patterns in how parents respond to children’s emotions have been found: parenting that is more individualistic/independence oriented, parenting that is primarily collectivist or relationally oriented, and parenting that is combination of individualistic and collectivist [23]. Each of these contribute to different dominant patterns in how parents socialize children around emotions.

Western parenting styles are underpinned by an individualistic or independence orientation, where families emphasize autonomy, personal choices, and wellbeing of the individual [24]. As a result, in some countries, such as the USA, high arousal emotions such as anger, which assist the individual in the achievement of their goals, are likely to be more valued and expressed [25]. In other individualistic countries such as in Scandinavia or Northern Europe, although the goal of autonomy is valued, expression of high arousal emotions is less common [26,27], in part because individualism may be less valued than in countries such as the USA [28,29]. Parenting styles in these countries are influenced by beliefs that children need to learn to cope with emotions alone (facilitated by a dismissive or disapproving response) or beliefs that giving support and guidance to help children understand and regulate their emotions is important (facilitated by emotion coaching) [30].

Collectivist-oriented parenting is seen in Asia, Africa, Central/South America, and other non-Western countries where families spend time instilling in their children the beliefs and values held by previous generations with the goal to maintain society’s harmony [31]. In collectivist cultures where interpersonal harmony is important, intense emotions are expected to be regulated within relationships [32,33]. Countries that value shame and embarrassment (i.e., China and Japan) are more likely to use these emotions to achieve collective harmony when compared with Western countries [32]. Chinese parents take a more ‘moralistic’ approach [34]; specifically, when dealing with emotions. They have been found more likely to use guilt, shame, and displays of disappointment to shape their children’s expression of emotions [35]. Chinese parents have also been found to use higher control and more emotional manipulation compared with parents from Western countries [36]. In collectivist cultures, children and adults are more likely to suppress emotion [37]. However, some studies have found differences in collectivist cultures, with parents who value individualism more likely to use emotion coaching, whereas those who hold collective goals more likely to be emotionally dismissive [14].

In non-Western cultures, authoritarian responses (characterized by firm control) are more valued, underpinned by a mixture of collective and individualistic orientations. In Iran, parents commonly use authoritarian discipline strategies of punishment, refer to social norms as ways of responding to child behavior [38,39], and tend to hold more dismissive attitudes towards emotions, when compared with other cultures [40]. In Turkey, a somewhat mixed individualistic and collective orientation with a family model of ‘emotional interdependence’ has been found. Children are raised in a context of maintaining close relationships with their families but are also socialized to develop autonomy from a young age [41]. Altan-Aytun et al. [42] found that mothers who demanded obedience (authoritarian parenting) were less likely to encourage emotional expression and more likely to use punitive and minimizing responses to emotions. In contrast, mothers’ who used inductive reasoning (authoritative parenting) tended to use more supportive ES responses.

These findings suggest cultures differ in parent ES. Eisenberg et al.’s (1998) model, based on studies from the USA, outlines how optimal ES consists of parents being low in emotion dismissing and high in emotion coaching. Given that interventions can reflect the cultures of their developers, it is important to investigate if the intervention is culturally appropriate, effective, or even desirable in collectivist or mixed individualistic/collectivist cultures. Although TIK aims to change ES to align with ‘optimal ES’, as proposed in Eisenberg’s model, it is not known if this would be accepted as an ‘optimal ES’ by parents in different cultures. This study investigated if the delivery of the ideas, concepts, and content taught in the TIK group-parenting program were culturally appropriate and acceptable in Germany, Turkey, Iran, and China.

### 1.3. Parent Emotion Socialization Intervention: Tuning in to Kids

Group delivery of parenting interventions is one of the main ways of preventing or treating child mental health difficulties, supported by considerable evidence [43,44]. TIK is an evidence-based parenting program that targets ES to improve child emotion competence and behavior. Although the program has been disseminated in Australia and has demonstrated evidence in improving ES and children’s emotional competence, and reducing internalizing and externalizing problems [4,5,6], it is important to examine if the program is culturally appropriate in other countries and the cultural modifications that may be required.

### 1.4. Cultural Adaptation of Parenting Interventions

In recent years, using principles from implementation science, the process of adaptation and evaluation of the TIK program in different cultural contexts has begun. Whether a program is adaptable, acceptable, appropriate, and feasible is of interest to program developers and researchers, and those engaged in applying interventions from one cultural context to the next [45,46]. Bernal and Adames [47] highlight that care must be taken in finding a balance between maintaining fidelity of an intervention with allowing adaptations for ‘cultural fit’. Specifically, imposing the dominant cultural norms, values, and world-views of the program developers onto different ethnocultural groups should be avoided, and instead careful evaluation of the cultural appropriateness and acceptability of the intervention should be sought.

Developers of evidence-based parenting interventions have used the Ecological Validity Model (EVM; [48]) to explore whether an intervention is compatible with the cultural expectations and values of the populations where the intervention is applied. This model highlights a focus on change in language, people involved in the study, content, and goals, and how these impact on the adaptation process. Simply translating an intervention word for word is not sufficient for cultural adaptation due to different metaphors and idiomatic expressions across languages. Moreover, literal translations of wording can result in poorly constructed sentences, and mistranslations can be an outcome of translators not understanding sufficiently the original content (and meaning). Consequently, items may need to be altered or deleted, or new items may need to be generated [49]. For example, languages vary in their breadth of terminology for describing emotions, including the number of terms across the emotional continuum. Soriano and Ogarkova [50] report that English has as many as 2000 ‘emotion terms’, whereas Taiwanese Chinese is reported to have only 750. Among the Eskimos and Yorubas, there is no word for ‘anxiety’ and for Chinese speakers there is also no equivalent word [51]. Such differences may be due to broader cultural experiences—not just within the emotional realm. Inuit Indigenous people have 50 words for snow, compared to one in English [52].

The Cultural Adaptation Process (CAP; [53]) model has sometimes been used alongside the EVM. The model is based on the idea of involving important people in the process of adapting the intervention, such as treatment developers and cultural adaptation specialists, in collaboration with local communities. Turner [54] provides principles for guiding cultural adaptations so that they are sensitive to the influence of cultural family risks, level of acculturation, family migration, levels of trauma, family financial stressors, language preferences, and unique factors within the culture. These differences mean interventions may not always fit across cultures. Adaptation needs to also consider issues of history and trust of service delivery institutions, to ensure the intervention will be accessed by those for whom it is intended. This may be important in countries where interventions and institutions may not be trusted.

### 1.5. Aims

The current paper explored the delivery of an ES parenting program, TIK, in four countries (Germany, Turkey, Iran, and China) that have different parenting styles (individualistic, individualistic/collectivist, and collectivist) and different cultural beliefs and practices around expression of emotion. The aim was to examine content and delivery processes to identify changes necessary for cultural appropriateness of the program.

## 2. Materials and Methods

This study included data from four sites that had conducted randomized controlled pilot studies of TIK in Germany, Turkey, Iran, and China during 2015–2019. Researchers from each site had previously contacted the first or last authors wishing to adapt and pilot the TIK program in their countries. The current study used an online qualitative questionnaire completed in 2020 by the researcher team in each site (*n* = 4) to ascertain the processes used to adapt and translate the program. This questionnaire was developed by the first, second, and last authors by reviewing the literature about adaption of programs to different cultures, use of the Ecological Validity Model and the Cultural Adaptation Process model to generate key questions [48], and a process of triangulation to reduce bias [52]. The questionnaire used open-ended questions in which researchers in each site were asked to describe: (1) what adaptations they made to the program; (2) how well the concepts in TIK were able to be translated; (3) who delivered the program; (4) what the diversity of cultures in their participant mix were; (5) whether changes were made to metaphors/idiomatic expressions to fit cultural expectations; (6) whether cultural modifications were made to fit the program to cultural expectations, in addition to the material included/excluded; (7) how much change was needed to concepts; (8) whether changes were made to the delivery/methods; (9) how much further change was recommended to the program following the pilot; (10) whether the intervention was feasible; (11) how accessible the program resources were; (12) how effective they perceived the program was for the site; (13) how well the program fit with their culture and what further adaptations were needed to make the program culturally appropriate; (14) how accepting parents were to the ideas; and (15) what the perceived barriers to delivery of the program were.

Once collected, data were organized in Excel and then thematic analysis was used to analyze the data following the process outlined by Braun and Clarke [55] involving: (1) becoming familiar with the data; (2) generating initial codes; (3) searching for themes; (4) reviewing themes; (5) defining and naming themes; and (6) producing the report. Once themes were identified, each site was asked to review the themes for triangulation. The four sites were also asked for additional reflection, in order to provide further explanation and detail about their site. A descriptive narrative format was chosen to report the findings.

### 2.1. Reflexivity and the Researcher-Participant Relationship

Transparency is an important part of conducting qualitative research and the current study sought to examine the role of the researchers in creating the measure used and evaluating the data [56,57]. The first, second, and last authors (a New Zealand PhD/clinical psychologist with 23 years’ experience developing, evaluating, and disseminating TIK; a Hong Kong Chinese undergraduate psychology student; and a German PhD psychologist with 19 years’ experience in researching emotion socialization, and in developing, delivering and evaluating versions of TIK) who created the questionnaire, and selected and analyzed the themes, were influenced by their own cultural experiences with emotions, experiences of parenting, and their extensive knowledge of the TIK program when interpreting the data. This was especially true for the first and last authors, who are deeply immersed in the TIK suite of programs and highly value the use of emotion coaching to help children understand and regulate emotions. This may have biased their selection of themes and interpretation of the adaptation processes across the sites (while also aiding this analysis). Further, a number of the researchers in the different sites were graduate students, and this may have created a power imbalance with the first/last authors who are the program developers. Site researchers were also included as study authors.

### 2.2. Participants

In all sites, the program targeted mothers of 4- to 6-year-old children attending kindergarten. Participants in all sites were recruited in kindergartens through advertising posters and letters sent home to parents inviting them to attend a parenting program and participate in a research study. In each site, parents self-selected whether they wished to enroll, and all gave informed consent. The inclusion criterion was that parents had sufficient comprehension to understand the materials and complete questionnaires in their language. The exclusion criterion was a diagnosis in the child of Intellectual Disability or Autism Spectrum Disorder. In Iran, children were selected who had elevated anxiety or behavior problems on the Achenbach Child Behavior Checklist. In all sites, 100% of parents completed the program (4+ sessions).

Mothers’ education and working status varied within and across sites. In the Turkish sample (*n* = 59), mothers’ education was from high school to postgraduate degree, 35% were working, with 65% at home. In the German sample (*n* = 54), 42% had a university degree, 23% had finished high school, and 30% had secondary school/vocational training. Most were working (78%), with 6% studying and 14% at home. In the Iranian sample (two separate samples; *n* = 56; *n* = 54), education varied from high school (25%) to university (75%); however, 87% were at home. In the Chinese sample (*n* = 89), 97% had a university degree.

Further information about parents from each site is not included here because data collection for this study was from the site adaptors in each country rather than from parents.

### 2.3. Tuning in to Kids (TIK) Intervention

The TIK program is a 6 × 2 h group-session parenting program. Based on theories of ES [2], mindfulness [58], and emotion-focused therapy [59,60], the program teaches parents to use emotion coaching to assist children to learn emotional competence [16]. Parents learn the 5 steps of emotion coaching and from session 1 onward each session aims to develop parents’ use of these skills in different contexts. Session 1 introduces all the main concepts and builds group cohesion; session 2 has a focus on naming emotions; session 3 focuses on empathy; session 4 focuses on responding to children’s fears and worries and parent self-care; session 5 focuses on anger; and session 6 addresses sibling conflict and material not previously covered, consolidates what has been learned, and addresses plans for how parents can continue to use and develop their skills after the program finishes. Parents are taught skills to identify, understand, and regulate their own emotions throughout the program. They explore the influence of culture, family of origin, and their meta-emotion beliefs (i.e., thoughts and feelings about emotions) to understand their automatic reactions to emotions. The program uses psychoeducation, DVDs, role plays (scripted/unscripted; demonstration, fishbowl), paired exercises, group discussion, and home-activities to practice. A structured facilitator manual was used with translated parent handouts. An emotion awareness activity uses Bear Cards or Stickers that depict emotions to enable discussion. A list of parenting books that were complementary to TIK was provided in each site. DVD material from TIK was used in China with subtitles, used minimally in Turkey and Iran in English (with site researchers translating), and was not used in Germany. All groups, containing between 6 and 12 mothers, were held during the daytime.

In each site, TIK was delivered by Masters/PhD psychology students (site researchers) trained by the first and last authors. The Turkish adaptor spent time with the TIK team in Melbourne on a fellowship where she attended a two-day face to face training, co-facilitated a TIK program, and was supervised by the first author in adaptation and delivery. The German researchers attended online training, and the translation and delivery was supervised by the last author, a native German speaker. The Iranian researchers received online training and supervision with the first author. The Chinese adaptor attended a face-to-face training with the last author but did not have supervision during delivery. Consistent with the Cultural Adaptation Process (CAP; [53]) model, the site researchers, who all had a high level of interest and motivation in the program, were central to the adaptation process.

### 2.4. Ethics

Each site had ethical approval from the respective university human ethics committees to conduct the research.

## 3. Results

All sites emphasized that the concepts in the original TIK program were the main focus of delivery; however, a number of themes emerged in the data about adaptations that were required, translations and language that was used, content additions, and adjustments to enable cultural appropriateness. The results are organized under these main themes outlined in Table 1 that were generated from the survey of site researchers, about the TIK program.

### 3.1. Adaptations Required for each Site

A number of adaptations were reported by each site, including to the way the key concepts were delivered. These sub-themes included: extra time for explaining key concepts; modification of examples and parent handouts to make them culturally specific; providing further explanations about basic parenting concepts (e.g., the role of parenting, how to read books to children); varying methods of conducting role plays (preferences for either scripted, fishbowl, or unscripted role plays); building up slowly to the use of self-disclosure across sessions to enable trust; and more time to address and shift parents’ beliefs about the causes of children’s emotions (e.g., that children’s expression of emotions were manipulative, considering emotions underlying behavior). These sub-themes are described across each site beginning with Germany, a more individualistic culture, then Turkey where there is a mix of individualistic and collectivistic practices, followed by Iran and China, which are more collectivistic cultures.

In Germany, exercises that required emotional openness were often not liked by participants. Initial feasibility testing suggested that exercises requiring ‘revealing secrets’ led to parents’ remaining silent and not being comfortable sharing. For example, in the first session the usual warmup involves parents speaking in pairs about ‘What is one thing we would never guess about you?’. The site researchers removed this question, staying with the less personal questions, “what did you eat for breakfast at age 4?” Paired exercises for practicing and exploring individual experiences were often preferred by parents, rather than group discussion. German mothers often viewed children’s emotions as manipulative, and this had to be addressed slowly. Additionally, the German site had to establish group guidelines for how parents would address each other during the sessions using either ‘Du’ (informal) or ‘Sie’ (Formal). Typically, ‘Du’ was chosen due to conversations about emotions being more intimate. Germans did not like the idea of keeping an ‘emotion diary’ and felt it was too personal therefore, facilitators asked parents to provide ‘summaries’ to record their interactions. During scripted role plays, parents often ‘left their role’ and discussed the situation at a meta-level criticizing the interaction as unrealistic. These parents preferred to use their own words instead of the scripted role plays. When it came to unscripted role plays, some parents found these helpful, while others were concerned about not being able to master the task. These problems led facilitators to implement an adaptive, flexible approach. Whenever necessary, facilitators used scripted or fishbowl role plays and, in later sessions, participants chose between scripted, fishbowl, or unscripted role plays to enhance self-determination. In paired work, parents who mastered the technique of emotion coaching and parents who were struggling were paired up. Discussions during the role play had to be consistently prevented by the facilitators until the reflection phase.

In Turkey, the main concepts of TIK were appropriate, however, greater explanation was needed for some concepts and emotion language. Focusing on exploring children’s emotions and not just behaviors was a novel idea. Mothers were reportedly open to supportive ES skills and wanted ways of teaching their children about emotions to relieve distress. The idea of low- and high-intensity emotion was not fully understood and so these differences were described using the associated body language and facial expression. Changes were made to handouts to ensure cultural appropriateness (e.g., swapping ‘Santa’ to an imaginary hero; removing reference to a garden because these are uncommon in Turkey). Parents needed more information about how to read books with their children that scaffolded emotion conversations and how to help children draw pictures of their emotions. Greater explanation for Step 5 of emotion coaching—the problem-solving step—was needed since the general tendency of parents was to eliminate the problem rather than assist their child to solve the problem. Dismissing positive emotions was common among parents (such as to dismiss pride in creating a colorful picture). To help parents empathize with their children, the adaptor used TIK exercises such as ‘Descriptive Praise’ or ‘The Emotion Detective’, where parents considered their own adult equivalent scenarios to events that trigger emotions in their children. The adaptor reported rarely using fishbowl role plays (where one person plays the child and the rest of the group first emotion dismiss then emotion coach) due to parents’ difficulties in spontaneously responding. Instead, the facilitator provided demonstration role plays, modeling the difference between emotion dismissing and coaching; then, parents used scripted role plays and, in later stages of the program, practiced using unscripted role plays.

In Iran, a number of adaptations were needed to help with teaching basic parenting skills and to address parents’ beliefs about children’s motives for being emotional. Iranian mothers often viewed children’s emotions as manipulative and did not believe in the legitimacy of children’s emotions. Emotion coaching was misunderstood as being permissive, letting the child be spoiled, and a fear of losing parental authority and control. For example, when emotion coaching was initially explained, one mother reported, ‘If we pay attention too much to the feelings of our children and start to follow their feelings, they will be like spoiled brats and they’ll abuse this situation to achieve their demands!’ Greater exploration of parents’ meta-emotion beliefs via examining family of origin experience was necessary to shift negative attitudes about emotion coaching. Mothers wanted more time and support to help them with empathy, talking about emotions, times for emotion coaching, development of an emotion vocabulary, and working out how to use the ‘emotion talk time’ (time with a child to listen to their experience). Mothers were reluctant to engage in role plays and found the scripted role plays superficial and not applicable in real life. After session 3 this frustration lessened; however, there was still resistance to unscripted role plays for some. Because of this resistance, the site researchers recommended additional sessions to allow for more reflection time and practice.

In China, modifications were required to assist parent engagement in the exercises and to help participants understand the TIK concepts. The adaptor reported that more time was needed for this because the concepts were new for participants. Discussion about emotion-related parenting practices, empathy, the importance of non-derogatory and non-critical parenting, setting family limits, and responding to children’s misbehavior required considerable explanation and exploration. Empathy, for example, is not a single word in Chinese and requires a longer phrase to explain the concept. Changes to the method of delivery (such as more paired than group work) was necessary for parent engagement. Mindfulness exercises were used in sessions 2–6 (not just sessions 3 and 4) to improve this aspect of parent functioning. The adaptor believed more activities and discussions in pairs were needed to enable parents to practice the skills because parents felt shy or unmotivated to share in front of the group. Anticipating parents’ reluctance to engage in unscripted role plays, the Chinese adaptor created additional scripted role plays which parents found useful.

### 3.2. Translation of the TIK Emotion Coaching Concepts and Emotion Language

The main themes reported with regard to translations were that sites needed to make changes to the program concept descriptions and define emotion terms used by the culture.

In Germany, there were very few difficulties reported by the site researchers with translating words and concepts in the program.

Some emotion expressions had no Turkish equivalent and required the adaptor to look at Turkish literature, liaise with Turkish emotion researchers, and then go back to the program developers to check the translated concepts were accurate. Fewer words for emotions were available in the Turkish emotion vocabulary, so alternative descriptors were used. For instance, the same Turkish word could be used to describe ‘frustration and disappointment’; ‘grumpy and angry’; ‘shy and embarrassed’. To identify the differences between these words, uncommon emotional expressions or some feeling states were used instead of emotion words (i.e., ‘utangaç’ for ‘shy’ and ‘mahçup’ for ‘embarrassment’). The concept of ‘tuning in’ was hard to translate and ended up being translated as ‘paying attention to children’s emotions’. Although there are a variety of emotion words in Turkish, parents’ emotional vocabularies were often limited by virtue of their own exposure to the terminology. The ‘List of Emotion Words’ handout helped differentiate emotional expressions such as ‘left out’, ‘outraged’, or ‘frustrated’. TIK includes a range of English words for ‘anger’ such as ‘mad’, ‘annoyed’, or ‘furious’. Due to fewer expressions in Turkey, feeling ‘irritated’ or ‘angry’ were most commonly used. Concepts of emotion competence were known in Turkey due to changing parenting trends emanating from social media. However, meta-emotion was a new concept and required substantial explanation for parents to understand. The adaptor used the ‘Bear Cards’ (pictures of bears with different emotion expressions) to assist when asking ‘How did you know this was the emotion you felt?’ and ‘How did you feel about feeling this emotion?’ This allowed exploration of the meta-emotion concept.

In Iran, although there is a wide range of emotion terms in Persian, the site researchers reported that parents had limited knowledge of these. For example, they reported there are a number of words for low-intensity anger; however, parents only tended to use two words—angry or irritated. (The English version of the ‘List of Emotion Words’ in the TIK program has 32 variations for anger). The site researchers believed this was due to a lack of awareness and acceptance of emotions rather than a lack of options in the Persian language. Mothers reported having difficulty accepting that some emotions were universal and normal. Religion played an important role in determining attitudes to some feelings. For example, envy in Islam is one of the seven great sins. When envy and jealousy were discussed by a parent or child, many mothers denied this emotion, or felt guilty or uncomfortable about it.

In China, English emotion words did not always translate easily because a single word was not available to capture the meaning. The site researchers created an additional handout to explain the emotion terms and that correct use depended on the context.

### 3.3. Content Additions

In some sites there needed to be some additions to the program. Turkey, China, and Iran all needed to address basic parenting strategies required to apply TIK content. Turkey encouraged parents to engage in interactive book reading with children and discussed how to respond to set simple limits. In Iran, the site researchers reported that material to assist in routines, boundaries, and praise would have been useful. They also reported parents needed guidance on how to help children sleep in their own beds. ‘Lots of parents hadn’t separated (from) their children at night and they needed to get strategies to deal with the separation problems. They don’t have adequate information about setting limits and boundaries’. In China, additional information was included about separation anxiety and sadness, and exploration of ‘tiger-mother’ protectiveness that was commonly used. Almost no additional content was needed in the German adaptation. One exception was one group wanted more input on setting limits for inappropriate behavior and use of logical consequences.

### 3.4. Cultural Appropriateness of the Program

With respect to the program fit with the culture, sites varied in their acceptance of the ideas depending on the wider attitudes to parenting and emotion.

In Germany, overall, the program was deemed culturally appropriate. Both Germany and Australia share values common to individualistic cultures. However, intense emotional experiences (e.g., to be overwhelmed in quarrels with one’s child) were less likely to be shared with other group members. Facilitators needed to carefully observe the reactions of the group members such as discomfort or criticism when parents engaged in greater emotional self-disclosure.

In Turkey, the site researchers indicated that the program fitted well because of current societal changes toward more individualism. The adaptor reported, ‘In collectivist family structures, expressing emotions is not accepted and most of the parents in (the) new generation grew up in this kind of family structure. Now those parents are in a transition age. They are transforming their family structures into a more individualistic type in which emotions are acceptable and important. In recent years, parenting behaviors are shaped according to the children’s needs and society’s structure. Therefore, emotions and their expressions have become important in my culture. Parents, especially mothers, increasingly share more positive emotions and show more emotional interactions with their children. Emotional competence has become a widely accepted concept among social and educational institutions. As a result, interventions raising children’s emotional competence and family meta-emotions are gaining importance in Turkish society’.

In Iran, the site researchers indicated that the program needed further modifications to enable a better fit with Iranian culture and parent’s needs, in addition to more sessions to consolidate the learnings. In the Iranian collectivistic culture, wider family involvement in parenting is common, and some mothers complained about the interference of their own mothers or their husband’s family in their parenting. Although the majority of mothers tried to be independent, to modify their parenting toward the individualistic values of paying attention to the child’s feelings and allowing them to express themselves, this resulted in some conflicts within their wider family. The site researchers also believed mothers needed additional sessions to address their meta-emotion beliefs that limited the application of some of the program content. For example, one mother fed back to the adaptor, ‘We ourselves do not know how to talk with our husbands when we are angry or feel sadness, so how can I teach it to my child? I think I need to know myself better to be able to help my child’.

In China, the site researchers reported that culturally it was not common to notice and talk about emotions, especially those related to the child’s individual needs. Consequently, the adaptor reported that Chinese parents were less proficient at responding to and accepting of emotions. They reported that, in recent years in Chinese society, emotion has begun to be noticed and parents have increasingly become aware of the importance of talking about emotions; however, they believed there was still a long way to go for Chinese parents to become comfortable with emotions and talking about them.

## 4. Discussion

This study was a preliminary qualitative evaluation of an emotion socialization group parenting program, Tuning in to Kids (TIK), in four countries (Germany, Turkey, Iran, and China) that have different parenting styles (individualistic, individualistic/collectivist, and collectivist). Specifically, the study aimed to examine cultural appropriateness of content and delivery processes to examine if adaptations of the program to each country required changes. Although the TIK parenting program was able to be delivered consistently with the original manual across the different sites, themes emerged in the different sites of what was needed to enable culturally appropriate delivery of the program in each country. These included challenges with adaptation of the program concepts; translations and use of emotion language; content additions in basic parenting concepts, and cultural fit. Each of these themes is discussed.

### 4.1. The Challenges with Adaptation of Program Concepts

Across the sites, there was a consistent theme that TIK program concepts presented a major cultural and conceptual challenge for parents. Each site required more time to assist parents with understanding the concepts taught in the program and in changing their beliefs about the benefit of talking about and responding to emotions. Many participants held beliefs that responding to emotions was ‘giving children attention’ and would make them more emotional and demanding. Sites reported that it was challenging to shift parents’ attitudes away from believing that their children’s emotional displays were manipulative, needed control, or were social transgressions, to instead valuing and accepting their children’s emotions. Shifting such deeply held beliefs requires motivation to change, which can come through understanding (i.e., the impact of parents’ responses to emotions on children’s subsequent emotional competence), acquiring skills to respond differently (such as use of the five steps of emotion coaching rather than emotion dismissing), and then having the opportunity to apply the skills and reflect on what worked and what did not (through role plays and then actual interactions with their children followed by further debriefing with the support of the facilitator and the group). This process of insight, skill development, and experiential application has been found to be integral to changing patterns of intergenerational neglect and abuse [61] and may be necessary to shift beliefs about emotions.

Role plays were a challenge for each site; however, they are a fundamental part of the TIK program because parents frequently have difficulty changing their responses to children’s emotions, especially when they or their child are emotional. Experience (i.e., being in the child’s role) and practice (i.e., being in the parent role) is essential for deeper understanding and preparing responses for emotional encounters at home. Many parents from the different sites resisted role plays despite the site researchers outlining how central these were to the learning. The way each site adjusted to the challenge of using role plays appears, in part, a reflection on their own culture. German parents wanted control over which type of role play they preferred and often were concerned about not getting the skills correct; Turkish parents wanted to see role plays demonstrated and understand them before slowly trying out the different exercises; Iranian parents wanted to discuss (and not do) the role plays or were critical of the scripted role plays but, over time, some parents who actively engaged in the program were better able to try out the unscripted role plays in the group setting; the Chinese adaptor predicted that parents would not tolerate unscripted role plays and created additional scripted ones. All adjustments were deemed culturally appropriate by the facilitators and still allowed parents to learn the key program concepts with high fidelity. The TIK program highlights that many parents understand and accept the ideas of emotion coaching, but it is only with practice (ideally first in the group setting) that the skills are acquired, and then parenting changes can occur. Role plays have been found to be important for building skills in different contexts such as for medical students learning communication skills [62] and clinical students learning therapeutic skills [63], and for acquiring new parenting skills [64]. Perhaps especially when emotions are likely to activate parents’ automatic responses of being emotionally dismissive or critical, learning through rehearsal is central for behavioral change and ensuring the effective implementation of an intervention [65].

Sharing personal experiences about emotions was not always comfortable for parents, especially when disclosing to the whole parenting group. In the German site, the researchers were required to accept reduced self-disclosure in the whole group in the early sessions. Paired activities, and less whole-group discussion, were also preferred in China because parents were often uncomfortable or less motivated to share with the larger group. In some Western countries, groups are used as an important method contributing to change, with benefits including a reduction in the experience of being alone through sharing; the role of support; clarity possibly being helped by modeling and learning through watching/hearing from others, reflecting, and expressing one’s own experiences; and allowing the non-confrontational exposure to new ideas [66]. However, this aspect of learning, involving sharing and disclosure in a public space, may have a strong cultural component that is more acceptable in Western, individualistic cultures [67,68]. Sharing of experiences was seen as less problematic in Iran and Turkey; however, the Turkish adaptor reported some parents were reluctant to attend the group in the first place because of a perception of shame that came with attending a parenting program, a finding echoed by the German researchers. Speaking about a topic as personal as emotions appears to have been a challenge in each site because sharing emotion-related experiences was uncommon. It may take generations for a culture to be comfortable speaking about emotions in public.

### 4.2. Cultural Appropriateness of the Program

This theme showed that sites varied in how well the concepts of the TIK program were deemed culturally appropriate for their culture and this appeared to depend on the degree to which the country held more individualistic versus collectivist beliefs. In Germany, the approach of responding to children’s emotions fitted most easily, with parents applying the ideas at home; however, as mentioned previously, sharing one’s emotions with group participants remained a barrier. This is consistent with previous research with German families that has found that German people are less emotionally expressive compared with families in other Western cultures [27] and tend to talk less openly about their emotions [66]. In Iran, more time was needed in the program to assist parents with exercises and experiences that would help shift their attitudes that children used emotions to get attention or manipulate their parents. In Turkey, parents were very interested in the TIK approach but there was some resistance to focusing on the individual’s emotions at the expense of the family or community. Çorapçı [69] reported that some Turkish mothers are aligned with the ‘emotion coaching’ ideas defined by Gottman et al. (1996); however, as reported by parents in the TIK groups, this depended on whether they held more progressive versus traditional values surrounding parenting. In the Chinese site, parents were motivated to promote their children’s emotional competence but within the cultural limits of ensuring respect and adherence to the wider cultural need for harmony [70].

Despite these challenges, all sites found ways of helping parents talk with their children about emotions that fitted their cultural and family values about emotion expression. Successful cross-cultural adaptations of interventions require changes to the program delivery and content that promote a closer fit with the cultural norms, and are responsive to parents’ unique needs but also retain the key ideas and messages of the program [71]. TIK allows parents to pick and choose the parts of the program that fit for them and their families, and this may have enhanced perception of cultural appropriateness. Exploration of meta-emotions allows content to be driven by participants’ own experiences and may naturally lead to culturally appropriate discussions. A number of sites were challenged by the concept of helping their child develop greater autonomy in working through emotions and in separating from the mother. The Iranian researchers said that parents needed assistance with setting limits with their children to enable them to use emotion coaching more effectively. Modification of the program may be required in countries where children remain reliant on their parents for longer. This highlights how adaptation of a program needs to fit with cultural values and practice to be acceptable [48].

Emotion coaching was challenging in the sites where parenting was more traditionally authoritarian and collectivist. In the Iranian and Chinese sites, emotion coaching was initially viewed by parents as giving in or being permissive. Parents had difficulty seeing that parenting did not require domination or obedience to enable children to behave pro-socially. TIK taught parents to look beneath behaviors at the emotions driving these (the metaphor of the iceberg is used—surface behaviors can reflect underlying emotions; emotion coaching seeks to address these to reduce difficult behaviors). This concept was challenging philosophically and culturally across the sites. Parents who held more individualist beliefs about emotions appeared more accepting of the TIK concepts. In contrast, those holding more collectivist beliefs appeared to value lower intensity emotions. For example, in the Iranian collectivist culture, some behaviors and, consequently, some emotions, were validated more than others. Being compliant and submissive was seen to have greater value in the family system. As a result, the more the child was silent and agreed with the parent’s limits, the more favorably the child was viewed. In contrast, anger and other emotions underlying oppositional behavior were less likely to be validated, and instead ignored or punished. Changing this way of responding was a cultural challenge faced by the site researchers. Where the TIK program concepts were more akin with individualistic parenting (German and Turkish sites), emotion coaching was more easily aligned with cultural and parenting values.

TIK aims to help parents develop skills in empathizing with their children, a skill that many Australian parents also struggle with. It requires regulating one’s emotions and using emotion awareness to imagine how one’s child is feeling and then communicating this. Across the sites, parents often had limited emotional awareness, making empathy and understanding the child’s experience difficult. Further, Turkey and Iran reported that parents needed additional assistance in managing their own reactions. Germany and China did not report this, perhaps because both cultures are less likely to express emotions even though they may experience emotional reactions. Turkish and Iranian cultures may express emotions more intensely and parents may be less regulated when parenting. Both ways of regulating emotions (over- and under-regulated emotional expression) may limit parents’ ability to connect and empathize with their children.

### 4.3. Translations of the Program and Emotion Language

Challenges with emotion vocabulary, in addition to cultural acceptance of expressing emotions and using emotion words, were major themes. German translations were relatively straightforward, with most emotion terms available. The other sites reported they often had to assist with more description of an emotion to assist parents with understanding the concept. Broadening emotion vocabularies to capture different levels of an emotion was a part of the adaptation in each site. The site researchers reported that the primary problem was that parents had limited exposure to or use of emotion language. The use of role plays, DVD material, and children’s storybooks were helpful tools to enable exposure to emotion language but also to create opportunities of talking with children about emotions.

Cultures vary in their acceptance and experience of emotions that may provide some explanation for the differences in emotion language across cultural groups in this study. Western cultures have been found to be more comfortable with high-intensity emotions, such as anger, and to experience these more often and more intensely [25]. Conversely, Eastern cultures experience low-arousal emotions more frequently than high-arousal emotions. Talking about emotions and accepting their expression was a core challenge for parents across different sites. Asking children to express anger was not comfortable for Chinese parents and may reflect that asserting one’s individualistic goals is less acceptable within the collectivist culture. The site researchers also reported that parents had fewer language terms to express anger. The emotion of envy or jealousy in Iran was less acceptable and therefore harder for parents to empathize with or talk about. Further, some Iranian mothers reported that naming emotions and letting their children express these (without punishing them) resulted in conflict with grandparents and other family members, further highlighting a tension between the traditional values of collectivist cultures and the newer generational practices that put words to emotion experience.

### 4.4. Limitations and Future Recommendations

The evaluation of the adaptation of TIK occurred after the program had been translated and the pilot trials in each site had occurred. Retrospective reflections can be a limitation in research and future studies would ideally collect this data prospectively throughout the adaptation process (e.g., by using discussion groups, a language/adaptation panel). All sites piloted the program with mothers with greater formal educational attainments; this may have meant that they held more progressive attitudes to emotions that may not be representative of the wider culture. The Iranian pilot studies recruited parents of children at risk for anxiety and behavior problems, meaning the sample of Iranian parents may have had more difficult circumstances than those of the other three sites. Future studies where the samples recruited represent a wider diversity of socioeconomic backgrounds and needs, and where qualitative data are collected using interviews with parents, would be ideal. Delivery of the TIK groups in each site was conducted by postgraduate clinical and research students who varied in age and experience. This may have affected the delivery of the groups and their perceptions of how parents responded to the content. Future research where the program is evaluated with professionals who work in parenting education and have greater experience with group dynamics would be useful. Lastly, this adaptation evaluation was led by two of the Australian TIK team (Havighurst and Kehoe), which may have biased the interpretation of the findings.

Future adaptations of the program would benefit from addressing some of the main themes outlined in this paper. These include: more time to deliver the concepts; creating role play examples that are culturally appropriate and using these gradually in a way that fits with the culture (but does not avoid doing role plays); inclusion of basic parenting concepts (if these are not usually known) about issues such as consistency and routines in parenting, and the importance of reading with a child; if appropriate, slowing the speed of self-disclosure across the sessions; one–two additional sessions to enable understanding of the emotion concepts and words, and about how to apply the emotion coaching skills; and more time to help parents learn to understand and regulate their own emotions.

## 5. Conclusions

This study reports on the preliminary evaluation of the TIK program when delivered to parents of preschoolers in Turkey, Germany, Iran, and China to examine cultural appropriateness in order to aid adaptation of the intervention to different countries. The themes emerging from these different sites were that the program delivery and content were able to be adapted for cultural fit. Similar to Western cultures that value emotion expression, other cultures also have an interest in educating parents in how to support their children’s emotional development. The TIK methods of supporting emotion-related parenting were deemed culturally appropriate by the site researchers in individualist and collectivist cultures, and adjustments enabled the program to fit the cultural mores of their country. These adjustments were procedural changes to the methods, rather than alterations to the ‘theory of change’ or the philosophical assumptions of the program, and do not, therefore, affect the fidelity of the program [47]. These procedural adaptations also reflect the evolution of an intervention in keeping with the values and practices of any given culture. The next step will be evaluation using randomized controlled trials across these different settings to enable cross-cultural comparisons of outcomes.

## Figures and Tables

**Table 1 ijerph-19-10321-t001:** Themes of the main adjustments required across each site.

Main Theme	Germany	Turkey	Iran	China
Adaptations required to methods				
More time to explain concepts		X	X	X
New culturally appropriate examples		X	X	X
Explanation of basic parenting concepts		X	X	X
Adjustments to role plays	X	X	X	X
Slowing speed of self-disclosure	X			X
More sessions would have been useful			X	X
Challenges in translation				
Greater description needed of program concepts		X	X	X
Greater description needed of emotion words		X	X	X
Additional content needed	X	X	X	X
Adjustments for cultural appropriateness		X	X	X

## Data Availability

Not applicable.

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
