# Peer review of "A Preliminary Evaluation of the Cultural Appropriateness of the Tuning in to Kids Parenting Program in Germany, Turkey, Iran and China"

_ijerph, 2022, doi:10.3390/ijerph191610321_

Round 1

Reviewer 1 Report

Thank you for the opportunity to review this manuscript, which describes a qualitative assessment of the cultural appropriateness of a parenting program (i.e., TIK) in four countries. This manuscript makes a significant contribution to the literature on cultural adaptations of interventions and highlights the relevance of including culturally appropriate language, symbols, etc.

I have three minor comments I hope you will find useful:

1.      In section 4.2, page 13, you comment that Turkish and Iranian cultures may express emotions more intensely than in Germany and China, and that “parents are therefore less regulated when parenting. This may limit parents’ ability to be calm and empathize with their children”.

Following Gross´ model of emotion regulation, it could be argued that restricted emotional expression does not necessarily mean that the adult is calm and able to empathize with the child (or anyone else), but only that the adult is able to control and inhibit emotional expression. A German and/or Chinese parent may be furious and not willing to listen to the child yet show limited signs of anger due to cultural rules about emotional expression. Conversely, and Iranian and/or Turkish parent may be expressive (even too much according to German or Chinese standards) yet be able to empathize with the child. Due to cultural variability in emotional expression, it may not always indicate dysregulation.

2.       In the conclusion, you argue that “Adaptations may result in a somewhat different program than originally intended by the program developers but reflect the evolution of an intervention in keeping with the values and practices of any given culture”.

When discussing cultural adaptation of interventions, Bernal and Adames (2017) address the problem of fidelity. That is, how much can an intervention be adapted and changed without being turned it into something completely different. The authors argue that the procedural model can be easily changed, but that the propositional model (i.e., theory of change) and the philosophical assumptions underlying the intervention should be kept. You may want to address this issue due to the several changes the TIK program underwent in the four countries described, and also because you cite the Ecological Validity Model (from Bernal et al) and Cultural Adaptation Process Model (by Domenech-Rodriguez, who collaborates with Bernal) in the introduction.

Lastly, future studies on this could benefit from interviews to parents who participated in TIK groups. I understand this is not the focus of the current manuscript, but it could be informative in the future.

Author Response

Reviewer 1

In section 4.2, page 13, you comment that Turkish and Iranian cultures may express emotions more intensely than in Germany and China, and that “parents are therefore less regulated when parenting. This may limit parents’ ability to be calm and empathize with their children”.

Following Gross´ model of emotion regulation, it could be argued that restricted emotional expression does not necessarily mean that the adult is calm and able to empathize with the child (or anyone else), but only that the adult is able to control and inhibit emotional expression. A German and/or Chinese parent may be furious and not willing to listen to the child yet show limited signs of anger due to cultural rules about emotional expression. Conversely, and Iranian and/or Turkish parent may be expressive (even too much according to German or Chinese standards) yet be able to empathize with the child. Due to cultural variability in emotional expression, it may not always indicate dysregulation.

Our Response: This is an excellent point. We have adjusted this paragraph to reflect this. It now reads:

“Germany and China did not report this perhaps because both cultures are less likely to express emotions even though they may experience emotional reactions. Turkish and Iranian cultures may express emotions more intensely and parents may be less regulated when parenting. Both ways of regulating emotions (over- and under-regulated emotional expression) may limit parents’ ability to connect and empathize with their children.

  1. In the conclusion, you argue that “Adaptations may result in a somewhat different program than originally intended by the program developers but reflect the evolution of an intervention in keeping with the values and practices of any given culture”.

When discussing cultural adaptation of interventions, Bernal and Adames (2017) address the problem of fidelity. That is, how much can an intervention be adapted and changed without being turned it into something completely different. The authors argue that the procedural model can be easily changed, but that the propositional model (i.e., theory of change) and the philosophical assumptions underlying the intervention should be kept. You may want to address this issue due to the several changes the TIK program underwent in the four countries described, and also because you cite the Ecological Validity Model (from Bernal et al) and Cultural Adaptation Process Model (by Domenech-Rodriguez, who collaborates with Bernal) in the introduction.

Our response: Thank you – this review section was useful to look at. We have amended the conclusion to reflect this. It now reads:

“The TIK methods of supporting emotion-related parenting were deemed culturally appropriate by the site researchers in individualist and collectivist cultures and adjustments enabled the program to fit the cultural morays of their country. These adjustments were procedural changes to the methods, rather than alterations to the ‘theory of change’ or the philosophical assumptions of the program and do not, therefore, affect the fidelity of the program [70]. These procedural adaptations also reflect the evolution of an intervention in keeping with the values and practices of any given culture.”

Lastly, future studies on this could benefit from interviews to parents who participated in TIK groups. I understand this is not the focus of the current manuscript, but it could be informative in the future.

Our response: Thank you for this suggestion. We have extended the Limitations section to now be 'Limitations and Future Recommendations' and included a statement that addresses this point:

“Future studies where the samples recruited represent a wider diversity of socioeconomic backgrounds and needs, and where qualitative data is collected using interviews with parents would be ideal. ”

Reviewer 2 Report

The article, in my opinion, is very well elaborated, has a solid conceptual foundation, is methodologically well designed and the report of results is very consistent. It also constitutes a relevant contribution to the cross-cultural validation of intervention programs with families, of a preventive nature for the development of behavioral problems in childhood and the emotional well-being of the following generations. I can only make three comments, two minor suggestions for improvement and one substantive concern.

·       I believe it is convenient to use the technical concept of "Triangulation" to name the procedure of validation of results described in materials and method (lines 222-223).

·       The description of the participating mothers in the 4 countries is not systematic, I think it is necessary to use a table that better presents the characteristics between rows 257 and 264, but presenting the groups in an equivalent way between the countries including the data of studies and work but also others such as number of children and ages of the mothers to better contextualize the results.

·    The argument that lower emotional competence is associated with more behavioral problems (citation [20], lines 76-77) is very important to validate the need to implement TIK in countries with different cultures, however, the reference is from a Netherlandish study that has individualistic parenting patterns. This association does not necessarily occur in collectivist cultures, so it would be necessary to substantiate why the result obtained with TIK is still necessary in those countries. This point is very important, because if TIK were to be implemented systematically in collectivist countries (public policy) in no more than two generations, very important cultural changes would be produced that should be socially validated as desirable for local cultures. This discussion is not made in the article, it is assumed that if it has good results in its origin (Australia) it should also be good in other countries, but this is not necessarily the case.

Author Response

Reviewer 2:

I believe it is convenient to use the technical concept of "Triangulation" to name the procedure of validation of results described in materials and method (lines 222-223).

Our response: Thank you – we have amended the wording of ‘transparency’ to ‘triangulation. We had introduced the concept of triangulation on the previous page so it is not necessary to define this again.

The description of the participating mothers in the 4 countries is not systematic, I think it is necessary to use a table that better presents the characteristics between rows 257 and 264, but presenting the groups in an equivalent way between the countries including the data of studies and work but also others such as number of children and ages of the mothers to better contextualize the results.

Our response: Because this study is an evaluation of the experience of program site adaptors rather than inclusion of parents as participants, we do not believe it is appropriate to include more detail of the participants in each pilot site. We do not report on data from parents. However, to clarify this, we have added a statement that for the purposes of this study.

 “Further information about parents from each site is not included here because data collection for this study was from the site adaptors in each country rather than from parents.”

The argument that lower emotional competence is associated with more behavioral problems (citation [20], lines 76-77) is very important to validate the need to implement TIK in countries with different cultures, however, the reference is from a Netherlandish study that has individualistic parenting patterns. This association does not necessarily occur in collectivist cultures, so it would be necessary to substantiate why the result obtained with TIK is still necessary in those countries. This point is very important, because if TIK were to be implemented systematically in collectivist countries (public policy) in no more than two generations, very important cultural changes would be produced that should be socially validated as desirable for local cultures. This discussion is not made in the article, it is assumed that if it has good results in its origin (Australia) it should also be good in other countries, but this is not necessarily the case.

Our response: Emotional competence in children is associated with a range of developmental outcomes across cultures. We have added a statement to ensure this link is clear, hence justifying an emotion competence intervention. We could give a more extensive review of this finding but this might not be necessary once we include a statement that this finding occurs across cultures.

We have changed the wording to be: “From early in life, children learn skills in understanding and managing their emotions, known as emotional competence [3]. These skills influence their social relationships, behavior, capacity to direct attention and learn, and affect how they cope with stressful events and transitions, a finding seen across cultures (ibid).”

Reviewer 3 Report

The manuscript presents an interesting study that explores the appropriateness to various cultural contexts of Turning in to kids (TIK), an empirically based parenting program that has been mainly developed and validated in the Australian community.

Recent literature (e.g., Helander et al., 2022) has highlighted the important role of parenting programs for prevention and treatment of child psychopathology. In this line, TIK is an intervention proposal based on the theory of emotional socialization that has proven to be effective for early prevention of emotional and behavioral difficulties. Large-scale dissemination will allow ethnically and culturally distinct communities to benefit from its positive effects on children's emotional development.

The study examines the delivery of TIK in four culturally different countries particularly heterogeneous in parenting styles and socially desired ways of emotional expression, thus aiming to verify to what extent the intervention is compatible with the cultural traits of the target populations, as established by the Ecological Validity Model  (EVM; Bernal et al., 1995). This way, the authors implicitly acknowledge that evidence-based parenting interventions, in the words of Bernal & Adames (2017), “contain values, norms, beliefs, and worldviews that may be contrary to those by many ethnocultural groups” and, therefore,  cultural pluralism should be considered in dissemination research. Overall, with this study, the authors are contributing to the current challenge of designing effective science-based prevention programs that are also culturally pertinent.

For all these reasons, this paper is relevant for the advancement of research on the cultural adaptation of interventions, an area that, being relatively new in prevention science, is highly significative for the optimal transfer of intervention technology from research to the practice setting. In summary, I consider that the topic addressed in this paper is of interest to IJERPH readers.

Next, I propose the following suggestions that would favor the clarity of the writing and optimize the readability of the study:

Although the content of the introduction is adequate, as it addresses the main aspects that justify the relevance of the study, I recommend the inclusion, in subsection 1.4 (page 4), of a brief mention to one of the main dilemmas faced by cultural adaptations of prevention interventions: i.e., the balance between fidelity in the implementation and the necessary flexibility for an effective adjustment to the cultural values ​​that need to be respected by the intervention.

This issue is treated, for example, in the special monograph of Prevention Science (2017): "Challenges to the Dissemination and Implementation of Evidence Based Prevention Interventions for Diverse Populations”. The experiences and conclusions collected in several papers regarding adaptations to collectivist populations (as is the case of Latino communities) can serve as a reference for the authors (e.g., Bernal and Adames, 2017; Castro et al., 2017).

Likewise, I recommend removing the explanation of why the simple linguistic translation is insufficient for a cultural adaptation (paragraph 1 on page 4), an advice largely assumed in current research, in favor of a further clarification of how the study follows the principles of EVM and CAP.

2. Regarding the organization of the results, I suggest that the authors consider the possibility of structuring them according to the type of parenting styles of the target communities (individualistic, collectivist and individualistic/collectivist), and not according to the countries. This alternative distribution will help to optimize the presentation of the results, as well as their connection with the discussion (note that, at this time, there is no correspondence between the layout of the subsections of both sections).

3. Given the breadth of the results of the study, and, to make them clearer, it is also recommended to include a summary table/s of the main information; highlighting, for example, the major difficulties in program application according with the three educational styles.

4. I also suggest extending the “limitations” section and setting out some future recommendations.

4.1. Regarding the limitations, it should be noted that not having used any prospective adaptation strategies (e.g., discussion groups, language translation/adaptation panel) may be directly related to the obstacles faced in the different applications of the program. On the other hand, the profile of the program deliverers may be related to the setbacks in handling certain unexpected situations arisen in the groups of mothers (e.g., related to processes of social desirability or, on the contrary, of rejection/resistance or demotivation among the participants). On the cultural adaptation of interventions, the ideal deliverer profile seems to be a person with knowledge of the sociocultural norms of the targeted context and experienced in managing group dynamics (e.g., community agents).

Regarding future recommendations, the authors are suggested to reflect on some instructions that could serve as "program adaptation guidelines" (e.g., in terms of flexibility of the content and/or the training methodology) to maximize cultural adjustment.

Author Response

Reviewer 3

Although the content of the introduction is adequate, as it addresses the main aspects that justify the relevance of the study, I recommend the inclusion, in subsection 1.4 (page 4), of a brief mention to one of the main dilemmas faced by cultural adaptations of prevention interventions: i.e., the balance between fidelity in the implementation and the necessary flexibility for an effective adjustment to the cultural values ​​that need to be respected by the intervention.

This issue is treated, for example, in the special monograph of Prevention Science (2017): "Challenges to the Dissemination and Implementation of Evidence Based Prevention Interventions for Diverse Populations”. The experiences and conclusions collected in several papers regarding adaptations to collectivist populations (as is the case of Latino communities) can serve as a reference for the authors (e.g., Bernal and Adames, 2017; Castro et al., 2017).

Our response:

Thank you for this suggestion. We have added reference to this body of work in the introduction and conclusion. In the introduction we have included the statement that:

“Bernal and Adames [47] highlight that care must be taken in finding a balance between maintaining fidelity of an intervention with allowing adaptations for ‘cultural fit’. Specifically, imposing the dominant cultural norms, values and world-views of the program developers onto different ethnocultural groups should be avoided, and instead careful evaluation of the cultural appropriateness and acceptability of the intervention sought.”

We have also added a statement to this end in the conclusion:

“The TIK methods of supporting emotion-related parenting were deemed culturally appropriate by the site researchers in individualist and collectivist cultures and adjustments enabled the program to fit the cultural morays of their country. These adjustments were procedural changes to the methods, rather than alterations to the ‘theory of change’ or the philosophical assumptions of the program, and do not, therefore, affect the fidelity of the program [47]. These procedural adaptations also reflect the evolution of an intervention in keeping with the values and practices of any given culture.”

Likewise, I recommend removing the explanation of why the simple linguistic translation is insufficient for a cultural adaptation (paragraph 1 on page 4), an advice largely assumed in current research, in favor of a further clarification of how the study follows the principles of EVM and CAP.

Our response: We have retained this statement. Throughout the next sections we have returned to the EVM and CAP models to outline how we have met some of these guidelines (see section under Methods and Measures. We have also added an additional sentence under the TIK intervention to talk about how the researchers were key in the adaptation.

“Consistent with the Cultural Adaptation Process model [CAP; 53], the site researchers, who all had a high level of interest and motivation in the program, were central to the adaptation process.”

  1. Regarding the organization of the results, I suggest that the authors consider the possibility of structuring them according to the type of parenting styles of the target communities (individualistic, collectivist and individualistic/collectivist), and not according to the countries. This alternative distribution will help to optimize the presentation of the results, as well as their connection with the discussion (note that, at this time, there is no correspondence between the layout of the subsections of both sections).

Our response: Thank you this is a good suggestion. We have reordered the paper to always report on Germany first, then Turkey, followed by Iran and China. In this way we decided it was preferable to show from the country requiring least change (matching to the individualistic country of Australia), followed by Turkey that has a mix of individualistic and collectivistic culture, followed by Iran and China where collectivistic cultural practices are more common.

  1. Given the breadth of the results of the study, and, to make them clearer, it is also recommended to include a summary table/s of the main information; highlighting, for example, the major difficulties in program application according with the three educational styles.

Our Response: Thank you this is a good suggestion. We have included a table of the main adjustments for each site.

  1. I also suggest extending the “limitations” section and setting out some future recommendations.

Our response: Thank you this section has been extended to include additional limitations an to outline future recommendations.

This section now reads:

4.4 Limitations and Future Recommendations

“The evaluation of the adaptation of TIK occurred after the program had been translated and the pilot trials in each site had occurred. Retrospective reflections can be a limitation in research and future studies would ideally collect this data prospectively throughout the adaptation process (e.g., by using discussion groups, a language/adaptation panel). All sites piloted the program with mothers with greater formal educational attainments: this may have meant that they held more progressive attitudes to emotions that may not be representative of the wider culture. The Iranian pilot studies recruited parents of children at risk for anxiety and behavior problems meaning the sample of Iranian parents may have had more difficult circumstances than those of the other three sites. Future studies where the samples recruited represented a wider diversity of socioeconomic backgrounds and needs, and where qualitative data was collected using interviews with parents would be ideal. Delivery of the TIK groups in each site was conducted by postgraduate clinical and research students who varied in age and experience. This may have affected the delivery of the groups as well as their perceptions of how parents responded to the content. Future research where the program was evaluated with professionals who worked in parenting education with greater experience with group dynamics would useful. Lastly, this adaptation evaluation was led by two of the Australian TIK team (Havighurst and Kehoe), which may have biased the interpretation of the findings.”

4.1. Regarding the limitations, it should be noted that not having used any prospective adaptation strategies (e.g., discussion groups, language translation/adaptation panel) may be directly related to the obstacles faced in the different applications of the program. On the other hand, the profile of the program deliverers may be related to the setbacks in handling certain unexpected situations arisen in the groups of mothers (e.g., related to processes of social desirability or, on the contrary, of rejection/resistance or demotivation among the participants). On the cultural adaptation of interventions, the ideal deliverer profile seems to be a person with knowledge of the sociocultural norms of the targeted context and experienced in managing group dynamics (e.g., community agents).

Our response: Thank you – we have included these issues in the limitations/future recommendations section (as outlined in previous question).

Regarding future recommendations, the authors are suggested to reflect on some instructions that could serve as "program adaptation guidelines" (e.g., in terms of flexibility of the content and/or the training methodology) to maximize cultural adjustment.

Our response: Thank you for this suggestion. We have added an additional paragraph to address this:

“Future adaptations of the program would benefit from addressing some of the main themes outlined in this paper. These include: more time to deliver the concepts; creating role play examples that are culturally appropriate and using these gradually in a way that fits with the culture (but does not avoid doing role plays); inclusion of basic parenting concepts (if these are not usually known) about issues such as consistency and routines in parenting, the importance of reading with a child; if appropriate, slowing the speed of self-disclosure across the sessions; 1-2 additional sessions to enable understanding of the emotion concepts and words as well as how to apply the emotion coaching skills, and more time to help parents learn to understand and regulate their own emotions.”